# Preoperative MRI and LDH in women undergoing intra-abdominal surgery for fibroids: Effect on surgical route

**Annie Kim[1], Leslie Boyd[2], Nancy Ringel[3], Jessica Meyer[2], Genevieve Bennett[4], Veronica Lerner** [1]*

**1** Department of Obstetrics & Gynecology and Women's Health, Albert Einstein College of Medicine, Montefiore Medical Center, Bronx, New York, United States of America, **2** Department of Obstetrics & Gynecology, New York University Langone Health, New York, New York, United States of America, **3** Department of Obstetrics & Gynecology, MedStar Health, Washington, District of Columbia, United States of America, **4** Department of Radiology, New York University Langone Health, New York, New York, United States of America

* lernervt02@gmail.com

**Data Availability Statement:** All relevant data are within the paper and its Supporting information files.

## Abstract

### Introduction

Our institution implemented a preoperative protocol to identify high-risk cases for which power morcellation should be avoided.

### Material and methods

In this retrospective cohort study, an institutional protocol requiring preoperative Magnetic Resonance Imaging with diffusion-weighted imaging and serum Lactate Dehydrogenase levels was implemented. Chart review was performed including all women who underwent intra-abdominal surgery for symptomatic fibroids from 4/23/2013 to 4/23/2015.

### Results

A total of 1,085 women were included, 479 before and 606 after implementation of the Magnetic Resonance Imaging / Lactate Dehydrogenase protocol. The pre-protocol group had more post-menopausal women (4% vs. 2%, p = 0.022) and women using tamoxifen (2% vs. 0%, p = 0.022) than those in the post-protocol group, but baseline patient characteristics were otherwise similar between groups. Incidence of malignant pathological diagnoses did not change significantly over the time period in relation to protocol implementation. The rate of open surgery for both hysterectomy and myomectomy remained the same in the year preceding and the year following initiation of the protocol (open hysterectomy rate was 19% vs. 16% in pre- and post-protocol groups, respectively, P = 0.463, and open myomectomy rate was 10% vs. 9% rates in pre- and post-protocol groups, respectively, P = 0.776). There was a significant decrease in the use of power morcellation (66% in pre- and 50% in post-protocol cohorts, p<0.001) and an increased use of containment bags (1% in pre- and 19% in post-protocol cohort). When analyzing the subset of women who had abnormal Magnetic Resonance Imaging / and Lactate Dehydrogenase results, abnormal Magnetic Resonance

**Funding:** The authors received no specific funding for this work.

**Competing interests:** The authors have declared that no competing interests exist.

**Abbreviations:** AAGL, Elevating Gynecologic Surgery, founded as American Association of Gynecologic Laparoscopists; ACOG, American College of Obstetricians and Gynecologists; BMI, body mass index; FDA, Food and Drug Administration; LDH, lactate dehydrogenase; LMS, leiomyosacroma; MRI, Magnetic Resonanance Imaging.

Imaging results alone resulted in higher rates of open approach (65% for abnormal vs. 35% for normal). Similarly, a combination of abnormal Magnetic Resonance Imaging and Lactate Dehydrogenase tests resulted in higher rates of open approach (70% for abnormal and 17% for normal). Abnormal Lactate Dehydrogenase results alone did not influence route.

## Conclusions

Rates of MIS procedures were decreased for women with abnormal preoperative Magnetic Resonance Imaging results. False positive results appear to be one of the main drivers for the use of an open surgical route.

## Introduction

Leiomyomas are a common condition with 70–80% of women receiving the diagnosis during their lifetime [1]. It is estimated that approximately 433,621 hysterectomies and 34,000 myomectomies are performed annually in the United States making these surgeries a significant percentage of the total national operative volume [2, 3]. Most of these procedures can be performed via minimally invasive route, decreasing overall morbidity [4, 5]. However, in order to do so, an appropriate method for removal of bulky fibroids is required. Preoperative diagnosis of cancer in women presenting with abnormal uterine bleeding remains a challenge [6, 7]. Power morcellation was challenged by a recent 2014 U.S. Food and Drug Administration safety communication out of concerns for dissemination of occult leiomyosarcoma (LMS), as well as other cancers, pre-malignant and benign pathologies that could lead to inadvertent spread of malignant and pre-malignant cells throughout the abdominal cavity, and lead to conditions such as iatrogenic leiomyomatosis [2, 8]. Several professional societies, including AAGL (Elevating Gynecologic Surgery, founded as American Association of Gynecologic Laparoscopists) and American College of Obstetricians and Gynecologists (ACOG), subsequently followed up with practice guidelines and committee opinions to address this issue, along with updates in the literature [9, 10]. In the wake of the FDA advisory, a number of studies have reported rising laparotomy rates, which in turn are associated with increased surgical complications [11, 12]. Evidence has consistently supported minimally invasive approaches to myomectomy and hysterectomy to minimize perioperative morbidity [3, 10]. A recent decision analysis by Siedhoff et al. incorporating updated leiomyosarcoma incidence estimates continues to suggest that mortality rates are low following hysterectomy for presumed benign fibroids, and that a minimally invasive approach remains a safe option [13]. A large descriptive study analyzing national trends in hysterectomy surrounding the FDA safety communication found that after an initial reactive decrease in minimally invasive hysterectomies, the effect reversed one year later [14]. In an attempt to address the morcellation safety concerns regarding LMS while continuing to offer the benefits of minimally invasive options to women, a protocol was instituted in our hospital based on work by Goto el al, requiring pre-operative magnetic resonance imaging (MRI) and serum LDH (lactate dehydrogenase) enzyme levels to identify women at increased risk of LMS [15]. Our preceding study demonstrated that when this protocol was implemented, MRI alone had a very high negative predictive value; however, it was associated with a high false positive rate and a low positive predictive value [16]. We are now reporting on the impact of a preoperative protocol on institutional surgical practice patterns at the time of FDA safety communication, and the influence of MRI and LDH results on selection of surgical route and tissue extraction methods.

## Material and methods

We conducted a retrospective chart review of a convenience sample of women who underwent intra-abdominal surgery for symptomatic fibroids at New York University Langone Health (NYULH) from April 23, 2013 to April 23, 2015, one year prior and one year after protocol implementation. An institution-wide protocol requiring preoperative MRI with diffusion weighted imaging, serum LDH isoenzyme 3, and total LDH was implemented on April 23, 2014 to determine eligibility for uncontained power morcellation based on pre-operative assessment of LMS risk. Our preceding publication describes this protocol in detail [16]. We included women between the ages of 18–99 with symptomatic fibroid uterus who underwent intra-abdominal surgery for leiomyomas. This included myomectomies (laparoscopic, robotic and open approaches) and hysterectomies (both total and supracervical, with laparoscopic, robotic, vaginal, laparoscopically-assisted vaginal and open approaches). We excluded hystero-scopic myomectomies. Patients were screened by a review of the operating room schedule during the study period. Eligible patients' charts were reviewed to extract relevant data including: patient demographics, preoperative workup, type of surgery and method of tissue extraction. We reviewed our internal institutional gynecologic cancer database (created and maintained for the purposes of tumor board treatment planning) to confirm that all cases of LMS were captured in our study.

In brief, a patient was eligible for uncontained laparoscopic power morcellation if all three results (MRI, total LDH, and LH3) were normal. If one of the results was abnormal, the surgeon was restricted from using the power morcellator; clinical management, as well as the resultant mode of surgery, and method of specimen removal was left to the surgeon's discretion. Alexis® contained extraction system bags (Applied Medical) became available at the time of protocol implementation; other types of containment systems including those that allow contained laparoscopic morcellation, were not.

Operating room schedules were reviewed to identify eligible women during the study period, and charts were reviewed to extract relevant data, including patient demographics, pre-operative work-up, pathology results, type of surgery, and method of tissue extraction.

The outcome metrics for this report include:

1. The impact of the protocol on institutional surgical practice patterns (including route, type, and extraction method), and

2. The influence of MRI and LDH results on surgical route.

Fischer exact tests were used to compare binary demographic parameters, pre- and post-protocol surgical patterns, uterine pathology incidence, and protocol adherence; Mann Whitney tests were used to compare continuous variables. Statistical analysis was performed using SAS software version 9.3 (SAS Institute, Cary, NC).

### Ethical approval

We obtained approval from the NYU School of Medicine IRB (i15-01220) with a waiver of consent prior to data collection on June 15, 2015.

## Results

A total of 1,678 women undergoing intra-abdominal fibroid surgery (myomectomies and hysterectomies) were identified during the study period. Of these, a total of 593 women were excluded for having a benign non-leiomyoma indication for surgery or for a history of prior uterine malignant or pre-malignant uterine conditions (i.e., endometrial hyperplasia or

**Table 1. Selected demographic and clinical characteristics.**

| Characteristic | Pre-Protocol Group N = 479 | Post-Protocol Group N = 606 | P-value |
|---|---|---|---|
| Age (years) | 42 (23–70) | 43 (22–76) | 0.114 |
| BMI (kg/m2) | 27.4 (16–52) | 27.5 (17–53) | 0.893 |
| Race | | | |
| White | 154 (32) | 161 (27) | **0.044** |
| Black | 193 (40) | 231 (38) | 0.466 |
| Hispanic | 21 (4) | 47 (8) | **0.021** |
| Other | 50 (10) | 79 (13) | 0.153 |
| Unstated | 3 (1) | 10 (2) | 0.124 |
| Menopausal status | | | |
| Pre-menopausal | 442 (92) | 559 (92) | 1.000 |
| Post-menopausal | 19 (4) | 10 (2) | **0.022** |
| Unknown | 18 (4) | 37 (6) | 0.095 |
| Family History of Cancer | 142 (30) | 162 (27) | 0.307 |
| Personal History of Cancer | 25 (2) | 18 (3) | 0.062 |
| Tamoxifen Use | 10 (2) | 3 (0) | **0.022** |
| Pelvic Radiation | 0 (0) | 1 (0) | 1.000 |

BMI = Body Mass Index.

Age and BMI are listed as average (range). Other characteristics are listed as N (%).

*P*-values determined by exact Mann-Whitney tests (for BMI and age) and by Fisher exact tests (for the rest of the characteristics listed in this table).

endometrial cancer). A total of 1,085 women were included in the study, 479 women in the pre-protocol group and 606 in the post-protocol group. Patient demographics are shown in Table 1.

The pre-protocol group had more women who were white, post-menopausal, and on tamoxifen when compared to the women in the post-protocol group. The post-protocol group had more Hispanic women. No differences were noted in age, BMI, cancer history, or history of pelvic radiation. Details on MRI and LDH test accuracy were described in our preceding study; in brief, most abnormal MRI results in the post-protocol cohort was noted to be false-positive [16].

There were no differences between pre- and post-protocol groups in the incidence of leiomyomas, uterine malignancies or atypical uterine tumors. Only the incidence of benign non-leiomyoma diagnoses, including adenomyosis and endometrial polyps, as diagnosed on surgical pathology, differed between the two groups: 33% vs 56% in the pre- and post-protocol groups for open hysterectomies, and 36% vs 46% in MIS hysterectomies, respectively (p = 0.05 for both) (Table 2).

Since our cohorts were based on the timing of protocol initiation, we compared surgical routes and methods of tissue extraction in both pre- and post-protocol cohorts (Tables 3 and 4).

A significant increase in the use of containment bags for tissue extraction was seen in the post-protocol group, from 1% to 19% (P<0.001) (Table 4).

There was no significant difference between pre- and post-protocol groups with regards to the proportion of open versus minimally invasive route of surgery (14% vs. 12% open procedure rates in pre- and post-protocol groups, respectively, P = 0.406). Open hysterectomy rates (19% vs. 16% in pre- and post-protocol groups, respectively, P = 0.463) and open

**Table 2. Comparison of uterine pathologies and associated surgeries performed in pre- and post-protocol groups.**

| Surgical Route (Number; Percentage) | | Uterine Pathology | | | | | | | | |
|---|---|---|---|---|---|---|---|---|---|---|
| | | Benign Fibroid | Benign Non-fibroid[a] | Endometrial Hyperplasia | Leiomyosarcoma | Endometrial Adenocarcinoma | Other Gynecological Cancer[b] | Non-Gynecological Cancer[c] | Smooth Muscle Tumor Variant[d] | No Uterine Pathology |
| Hysterectomy | Open — Pre-Protocol (N = 40) | 38; 95% | **13; 33%** | 3; 8% | 0; 0% | 2; 5% | 1; 3% | 1; 3% | 1; 3% | 0; 0% |
| | Open — Post-Protocol (N = 41) | 38; 93% | **23; 56%** | 2; 5% | 0; 0% | 0; 0% | 0; 0% | 0; 0% | 3; 7% | 0; 0% |
| | MIS — Pre-Protocol (N = 176) | 160; 91% | **64; 36%** | 2; 1% | 0; 0% | 2; 1% | 1; 1% | 0; 0% | 7; 4% | 2; 1% |
| | MIS — Post-Protocol (N = 220) | 205; 93% | **102; 46%** | 3; 1% | 0; 0% | 1; 0% | 0; 0% | 0; 0% | 3; 1% | 0; 0% |
| Myomectomy | Open — Pre-Protocol (N = 25) | 24; 96% | 2; 8% | 0; 0% | 0; 0% | 0; 0% | 0; 0% | 0; 0% | 1; 4% | 0; 0% |
| | Open — Post-Protocol (N = 30) | 28; 93% | 3; 10% | 0; 0% | 2; 7% | 0; 0% | 0; 0% | 0; 0% | 1; 3% | 0; 0% |
| | MIS — Pre-Protocol (N = 238) | 226; 95% | 27; 11% | 0; 0% | 0; 0% | 0; 0% | 0; 0% | 0; 0% | 4; 2% | 1; 0% |
| | MIS — Post-Protocol (N = 315) | 299; 95% | 54; 17% | 0; 0% | 0; 0% | 0; 0% | 1; 0% | 0; 0% | 3; 1% | 0; 0% |

Data are summarized as N; %.

All Fisher's exact P-values for association equal to 1.0 except for the association between benign non-fibroids and pre/post protocol in open and MIS hysterectomies (p = 0.05 for both, in bold).

[a] "Benign Non-fibroid" includes adenomyosis, adenomyoma, endometrial polyp, chronic endometritis, denuded endometrium, polypoid adenomyoma, endometriosis, fibrosis, infarctive endometrium, endometrium with cystic degeneration, atrophic endometrium, focal compression atrophy, tubal metaplasia, glandular crowding, proliferative endometrium, foreign body giant cell reaction, and chronic serosal inflammation

[b] "Other Gynecological Cancer" includes aggressive angiomyxoma, mullerian adenocarcinoma, and endometrial stromal sarcoma

[c] "Non-Gynecological Cancer" includes metastatic breast adenocarcinoma

[d] "Smooth Muscle Tumor Variant" includes adenostromyoma, smooth muscle tumor of uncertain malignant potential, symplastic leiomyoma, cellular leiomyoma, adenomatoid tumor, mitotically active leiomyoma, lipoleiomyoma, leiomyoma with atypia, and mitotically active smooth muscle tumor with atypia

**Table 3. Rates of open and MIS surgical routes for hysterectomies and myomectomies combined in pre-and post-protocol cohorts.**

| Surgical route | Pre-protocol (N = 479) | Post-protocol (N = 606) |
|---|---|---|
| Open | 65 (14%) | 71 (12%) |
| MIS | 414 (86%) | 535 (88%) |

MIS-Minimally invasive surgery

P = 0.406, Fisher's exact test for association.

myomectomy rates (10% vs. 9% in pre- and post-protocol groups, respectively, P = 0.776) in either cohort did not differ when broken down by the type of surgery.

There was a significant decrease in the rate of power morcellation use over the study period: 66% pre-protocol vs. 50% post-protocol (P<0.001) (Table 4).

Similar trends with regard to decreased use of power morcellation and increased use of containment bags for tissue extraction were noted for both hysterectomies and myomectomies (Table 5).

In the post-protocol cohort of 606 cases, 358 women had both MRI and LDH performed prior to surgery, a protocol compliance rate of 59% (S1 Table). Out of the 358 women with full protocol testing, only six women had abnormal pathology (two had leiomyosarcoma, one endometrial adenocarcinoma, one other smooth muscle cancer, and two had smooth muscle tumor variants) [16]. Sub-group analysis was performed by MRI and LDH results with regard to subsequent surgical route. Of those 358 women who had both MRI and LDH completed preoperatively, 203 had both negative MRI and negative LDH results; in this group, only 3% of women had an open procedure. In contrast, in the 23 women with both abnormal MRI and abnormal LDH, 17% underwent open surgery (P = 0.01 when compared to the group that had both normal results), while the rest underwent a minimally invasive procedure. Furthermore, 132 women had discordant results. A combination of an abnormal MRI with normal LDH was significantly more likely to be associated with an open procedure (9.3%; P = 0.02 when compared to group with both normal results), while normal MRI and abnormal LDH combination did not show statistically significant increase in open route. Abnormal MRI alone, regardless

**Table 4. Rates of tissue extraction methods in patients who had MIS surgery in pre-and post-protocol cohorts.**

| Tissue extraction method | Pre-protocol (N = 414) | Post-protocol (N = 606) | P-value |
|---|---|---|---|
| Power morcellation | 318 (66%) | 301 (50%) | <0.001 |
| Mini-laparotomy with containment bag | 6 (1%) | 113 (19%) | <0.001 |
| Mini-laparotomy without containment bag | 15 (3%) | 23 (4%) | 0.406 |
| Vaginal with containment bag | 0 (0%) | 2 (0%) | 0.506 |
| Vaginal without containment bag | 58 (12%) | 88 (14%) | 0.282 |
| Unknown* | 17 (4%) | 8 (1%) | 0.023 |

All values are N (%) unless otherwise indicated.

MIS-Minimally invasive surgery.

* "Unknown"—cases in tissue extraction method was not stated in the operative report.

Vaginal without containment bag—includes intact and fragmented specimen removal.

Contained tissue extraction methods include "mini-laparotomy with containment bag" and "vaginal with containment bag."

Uncontained tissue extraction methods include "power morcellation," "mini-laparotomy without containment bag" and "vaginal without containment bag."

**Table 5. Comparison of surgeries, associated routes and tissue extraction techniques by type of surgery in pre- and post-protocol groups.**

| | Surgery Type | | | | | |
| | Hysterectomy | | | Myomectomy | | |
| Tissue Extraction Route | Pre-Protocol (N = 216) | Post-Protocol (N = 261) | P-value* | Pre-Protocol (N = 263) | Post-Protocol (N = 345) | P-value* |
| --- | --- | --- | --- | --- | --- | --- |
| **Open** | 40; 19% | 41; 16% | 0.463 | 25; 10% | 30; 9% | 0.776 |
| **Power morcellation** | 103; 48% | 90; 34% | **0.004** | 215; 81% | 211; 61% | **<0.001** |
| **Mini-laparotomy with containment bag** | 0; 0% | 22; 8% | **<0.001** | 6; 2% | 91; 26% | **<0.001** |
| **Mini-laparotomy without containment bag** | 7; 3% | 13; 5% | 0.370 | 8; 3% | 10; 3% | 1.000 |
| **Vaginal with containment bag** | 0; 0% | 2; 1% | 0.503 | 0; 0% | 0; 0% | 1.000 |
| **Vaginal without containment bag** [a] | 58; 27% | 87; 33% | 0.134 | 0; 0% | 1; 0% | 1.000 |
| **Unknown**[b] | 8; 4% | 6; 2% | 0.421 | 9; 3% | 2; 1% | 0.012 |

Data are summarized as N; %.

*Corresponds to Fisher's Exact test of association between pre and post-protocol and each extraction route.

[a] Includes intact and fragmented specimen removals

[b] "Unknown" refers to cases in which MIS tissue extraction method was not mentioned in the operative report.

Contained tissue extraction methods include "mini-laparotomy with containment bag" and "vaginal with containment bag."

Uncontained tissue extraction methods include "power morcellation," "mini-laparotomy without containment bag" and "vaginal without containment bag."

of LDH results, was associated with higher rates of open approach (13.4% when abnormal vs. 3.4% when normal, P = 0.001). Abnormal LDH results alone did not significantly influence route (P = 0.19). Subgroup size was too small to perform analysis by procedure type (myomectomy and hysterectomy) within each subgroup.

## Discussion

In this large retrospective study, we report the institutional impact of a protocol requiring preoperative MRI and LDH testing on subsequent surgical practice patterns. The proportion of minimally invasive surgery, for both hysterectomies and myomectomies, did not change significantly after implementation of the protocol, and the vast majority of cases were performed via a minimally invasive approach. A significant decrease in power morcellation use and increase in the use of containment bags for tissue extraction were noted across surgery types (hysterectomies and myomectomies). An interesting effect of discrepant MRI and LDH abnormal result combinations was noted as we found that only abnormal MRI results (most of which were falsely positive) increased rates of open surgery in that subgroup of women.

Unlike large retrospective studies which showed a decrease in the proportion of minimally invasive hysterectomies in the immediate time period following the FDA warning statement on the use of power morcellation, we did not observe a significant change in the route of surgery [12]. At our institution, the protocol was utilized in order to continue to "safely" permit continued power morcellation. This may have created an environment that cushioned the effect of this warning compared to other surgical centers. Also, as a center with a large referral base for providing minimally invasive surgical options, there remained a continued

expectation that these options would remain available. While the hospital in our study opted to create this protocol to allow both power morcellation for women who qualify and contained tissue extraction, surgeons and women were also responding to media coverage and major society guidelines which advocated for contained methods. As a result, the protocol alone cannot be held primarily accountable for our findings.

In terms of clinical applications, the effect of MRI and LDH results on the route of surgery is interesting from an implementation standpoint. Given that most abnormal MRI and LDH results in our study were false positives (only two LMS cases were found in the entire cohort), the impact of the high false positive rate–which was associated with open procedures—is of concern to clinicians. First, we found that an abnormal MRI result, regardless of LDH result, was associated with higher rates of an open approach. It may be that an abnormal MRI alone, regardless of LDH results, may be worrisome enough to steer women and providers alike away from an MIS route. The impact of these results on patient stress or referrals to gynecologic oncology remains unknown. Second, many surgeons chose to forgo this protocol and abnormal testing was ignored in significant numbers of cases. Due to the high false positive rate, the use of LDH and MRI should not be routinely used as a method to screen for LMS in women undergoing surgery for fibroids.

Several studies emphasize the importance of pre-operative evaluation for occult malignancy, particularly for higher risk women [17, 18]. Other work has explored patient factors, surgical, and occult cancer-related outcomes [13]. However, no imaging study or test has emerged as a useful adjunct to clinical evaluation, and the approach described in the paper by Goto et al. has not been accepted as ready for clinical implementation as it has not been replicated; our study further shows the negative implications of this protocol in the "real world" setting [9, 10, 15, 16]. Meanwhile, MRI has been established as an important pre-operative imaging study used for "surgical mapping" to select myomectomy patients who are good candidates for an MIS approach, evaluate for such other relevant conditions as adenomyosis, endometriosis during pre-operative planning, and to guide intraoperative localization of myomas [19]. As a result, clinicians are faced with a clinical conundrum when discussing abnormal MRI results with their patients, despite knowing that they are likely to be false positive results. Our study suggests that in this setting, clinicians are not able to "ignore" positive MRI results, leading to higher rates of laparotomy in cases that likely could have been performed minimally invasively. Mechanisms to improve the sensitivity of MRI in the detection of LMS remain an important research goal. Further investigation is needed to assess the long-term effects and the cost effectiveness of the preoperative work up presented in this study. It would also be important to examine its impact in different institutions and practice settings. Future studies could also examine if women who planned a minimally invasive myomectomy changed their minds and ended up with an open hysterectomy as a result of abnormal MRI and LDH findings. Further, it would be helpful for clinicians to know the false negative rate of MRI and LDH in identifying LMS on final pathology.

Our retrospective study design, single center location, and a relatively small number of women in the abnormal MRI/LHD subgroups limits our ability to generalize our findings. Moreover, although the prevalence of benign and malignant uterine pathologies did not differ between pre- and post-cohorts, it is possible that selection bias was present since containment bags became available in post-protocol group. Strengths of our study include an overall large cohort exposed to the protocol and a "real world experience" which unveils the difficulties in implementation of a new clinical protocol.

It is beyond the scope of this manuscript to address institutional administrative decision-making around the time of morcellation controversy. Future work could be done to address the relatively low rate of compliance with this protocol, which could lead to concerns with

selection bias. While only 59% of patients have undergone both MRI and LDH testing, our hospital did not have any other parameters for patient selection, and in the future, it might be of interest to study how each individual surgeon made decisions at the time of pre-operative work up. This would have broad implications and may allow for an improved method to increase uptake in behavioral changes. Our study also was not able to address nuanced factors that go into decision-making with regard to route for surgery and extraction technique used, such as surgeons' personal preferences, previous experiences, culture and traditions of the institution, fear of missing an LMS pre-operatively. These everyday challenges need to be addressed in future studies, including cost considerations. Finally, as scientific advancements are made in understanding epigenetic and genetic landscape of uterine myomas, we hope that our treatment options will expand beyond what is available as this time [20].

## Conclusion

Although earlier studies have suggested an overall decrease in minimally invasive hysterectomies in response to the FDA warning on power morcellation, there was no change in rates of minimally invasive hysterectomies and myomectomies at our institution during a similar time period. More studies are needed to describe effect of pre-operative MRI and LDH testing on surgical route, referral patterns to gynecologic oncology, and patient perceptions of how those results affected their experience. Further research is needed to guide practices that will consistently identify the highest risk patients preoperatively and minimize the impact of unexpected cancers while affording the appropriate patients the benefits of minimally invasive surgery.

## Supporting information

**S1 Table. Surgical route stratified by MRI/LDH results in post-protocol cases where both MRI and LDH test was done according to the protocol.**
(DOCX)

**S1 Raw data.**
(XLSX)

## Acknowledgments

This abstract has been selected for NON-ORAL POSTER presentation at the SGS 46th Annual Scientific Meeting in Jacksonville, Florida, March 29-April 1, 2020. ID# 3272611. Conference was cancelled due to COVID-19.

## Author Contributions

**Conceptualization:** Leslie Boyd, Jessica Meyer, Genevieve Bennett, Veronica Lerner.

**Data curation:** Nancy Ringel, Veronica Lerner.

**Formal analysis:** Annie Kim, Leslie Boyd, Genevieve Bennett, Veronica Lerner.

**Investigation:** Leslie Boyd, Veronica Lerner.

**Methodology:** Leslie Boyd, Veronica Lerner.

**Project administration:** Nancy Ringel, Jessica Meyer, Veronica Lerner.

**Software:** Jessica Meyer, Veronica Lerner.

**Supervision:** Veronica Lerner.

**Writing – original draft:** Annie Kim, Genevieve Bennett, Veronica Lerner.

**Writing – review & editing:** Annie Kim, Leslie Boyd, Nancy Ringel, Jessica Meyer, Genevieve Bennett, Veronica Lerner.

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
