## [Decision Letter · Decision Letter 0]

11 Jan 2021

PONE-D-20-37531

Preoperative MRI and LDH in women undergoing intra-abdominal surgery for fibroids: effect on surgical route.

PLOS ONE

Dear Dr.,

Thank you for submitting your manuscript to PLOS ONE. After careful consideration, we feel that it has merit but does not fully meet PLOS ONE’s publication criteria as it currently stands. Therefore, we invite you to submit a revised version of the manuscript that addresses the points raised during the review process.

We look forward to receiving your revised manuscript.

Kind regards,

Diego Raimondo

Academic Editor

PLOS ONE

Journal Requirements:

2.Please ensure that you include a title page within your main document. You should list all authors and all affiliations as per our author instructions and clearly indicate the corresponding author.

3.In your Data Availability statement, you have not specified where the minimal data set underlying the results described in your manuscript can be found. PLOS defines a study's minimal data set as the underlying data used to reach the conclusions drawn in the manuscript and any additional data required to replicate the reported study findings in their entirety. All PLOS journals require that the minimal data set be made fully available. For more information about our data policy, please see http://journals.plos.org/plosone/s/data-availability.

 "no"

"no"

6. Please include your tables as part of your main manuscript and remove the individual files. Please note that supplementary tables (should remain/ be uploaded) as separate "supporting information" files.

Reviewers' comments:

Reviewer's Responses to Questions

**Comments to the Author**

1. Is the manuscript technically sound, and do the data support the conclusions?

Reviewer #1: Partly

Reviewer #2: Partly

2. Has the statistical analysis been performed appropriately and rigorously? 

Reviewer #1: Yes

Reviewer #2: Yes

3. Have the authors made all data underlying the findings in their manuscript fully available?

Reviewer #1: Yes

Reviewer #2: Yes

4. Is the manuscript presented in an intelligible fashion and written in standard English?

Reviewer #1: Yes

Reviewer #2: Yes

5. Review Comments to the Author

Reviewer #1: I read with great interest the manuscript, which falls within the aim of this Journal. In my honest opinion, the topic is interesting enough to attract the readers’ attention. Nevertheless, authors should clarify some points and improve the discussion, as suggested below.

Authors should consider the following recommendations:

- Accumulating evidence suggests that some intrinsic abnormalities of the myometrium, abnormal myometrial receptors for estrogen, and hormonal changes or altered responses to ischemic damage during the menstrual period may be responsible for the initiation of (epi)genetic changes found in uterine myomas. I would stress this point of paramount importance, referring to: PMID: 28875276

- To date, the pre-operative diagnosis and post-operative management of uterine sarcomas represent still a challenge. I truly think that it would be necessary to add few lines about the risk of facing off of a misdiagnosed sarcoma during the management of uterine myomas and how to proceed (authors may refer to: PMID: 32497607; PMID: 33098396)

Reviewer #2: the authors report a single center experience on morcellation after adopting an internal protocol for suspicious myomas.

The paper is well written and clearly expose the problem.

However, there are some limitations that needs to be clarified:

Line 150 -162: please report the indication for hysterectomy for patients included in the study, has polyps been considered an indication for hysterectomy? Would have been useful to include only hysterectomies for fibroids

Line 178- the authors report a protocol compliance of 59% that actually create a bias in interpreting the results.

Line 179 - 191: The data reported here are actually duplicated in table number 6 that is difficult to be read, i suggest to remove the table

The pre and post protocol analysis has a limited validity because of the too low compliance to the protocol therefore this should be considered a snapshot of the surgeon's decisional process following the 2014 facts. Would be nice to see if there is trend in the last 6 years and how the patients that did no underwent MRI were treated.

The fact that only 59% of the patients underwent both MRI and LDH means that in the centre there are other parameters used to stratify the risk, please report the how the patients were selected for the screening.

The paper would benefit of a post operative care analysis such as Hb drop, hospitalization etc to understand the real impact of the morcellation issue on our patients health.

6. PLOS authors have the option to publish the peer review history of their article (what does this mean?). If published, this will include your full peer review and any attached files.

Reviewer #1: No

Reviewer #2: No

---

## [Decision Letter · Decision Letter 1]

27 Jan 2021

Preoperative MRI and LDH in women undergoing intra-abdominal surgery for fibroids: effect on surgical route.

PONE-D-20-37531R1

Dear Dr. Lerner,

We’re pleased to inform you that your manuscript has been judged scientifically suitable for publication and will be formally accepted for publication once it meets all outstanding technical requirements.

Kind regards,

Diego Raimondo

Academic Editor

PLOS ONE

Additional Editor Comments (optional):

Reviewers' comments:

Reviewer's Responses to Questions

**Comments to the Author**

1. If the authors have adequately addressed your comments raised in a previous round of review and you feel that this manuscript is now acceptable for publication, you may indicate that here to bypass the “Comments to the Author” section, enter your conflict of interest statement in the “Confidential to Editor” section, and submit your "Accept" recommendation.

Reviewer #1: All comments have been addressed

2. Is the manuscript technically sound, and do the data support the conclusions?

Reviewer #1: Yes

3. Has the statistical analysis been performed appropriately and rigorously? 

Reviewer #1: Yes

4. Have the authors made all data underlying the findings in their manuscript fully available?

Reviewer #1: Yes

5. Is the manuscript presented in an intelligible fashion and written in standard English?

Reviewer #1: Yes

6. Review Comments to the Author

Reviewer #1: I carefully evaluated the revised version of this manuscript.

Authors have performed the required changes, improving significantly the quality of the paper.

7. PLOS authors have the option to publish the peer review history of their article (what does this mean?). If published, this will include your full peer review and any attached files.

Reviewer #1: No

---

## [Editor Report · Acceptance letter]

29 Jan 2021

PONE-D-20-37531R1 

Preoperative MRI and LDH in women undergoing intra-abdominal surgery for fibroids: effect on surgical route. 

Dear Dr. Lerner:

I'm pleased to inform you that your manuscript has been deemed suitable for publication in PLOS ONE. Congratulations! Your manuscript is now with our production department. 

Kind regards, 

on behalf of

Dr. Diego Raimondo 

Academic Editor

PLOS ONE